# Rapid calcification propensity testing in blood using a temperature controlled microfluidic polymer chip

Julia Bavendiek[1], Philip Maurer[1¤], Steffen Gräber[2], Andreas Pasch[3], Werner Karl Schomburg[1], Willi Jahnen-Dechent[2]*

1 KEmikro, RWTH Aachen University, Aachen, Germany, 2 Biointerface Laboratory, Helmholtz-Institute for Biomedical Engineering, RWTH Aachen University Clinics, Aachen, Germany, 3 Calciscon AG, Nidau; Lindenhofspital, Bern, Switzerland; Institute for Physiology and Pathopysiology, Johannes Kepler University, Linz, Austria

¤ Current address: MERLN, Institute for Technology-Inspired Regenerative Medicine, Maastricht, The Netherlands

* willi.jahnen@rwth-aachen.de

**Data Availability Statement:** All relevant data are within the manuscript.

**Funding:** This work was supported by a fund of the Federal Ministry for economic affairs and energy of

## Abstract

Phosphate toxicity is a major threat to cardiovascular health in chronic kidney disease. It is associated with oxidative stress, inflammation and the accumulation of calcium phosphate commonly known as calcification in soft tissues leading to functional disorders of blood vessels. An improved calcification propensity test for the assessment of phosphate toxicity was developed, which measures the velocity of calcium phosphate mineralization from colloidal precursors *in vitro.* This so called T50 test measures the transformation from a primary into a secondary form of nanosized colloidal plasma protein-calcium phosphate particles known as calciprotein particles. The T50 test in its previous form required a temperature controlled nephelometer and several hours of continuous measurement, which precluded rapid bed side testing. We miniaturized the test using microfluidic polymer chips produced by ultrasonic hot embossing. A cartridge holder contained a laser diode for illumination, light dependent resistor for detection and a Peltier element for thermo control. Increasing the assay temperature from 37˚C to 75˚C reduced the T50 test time 36-fold from 381 ± 10 min at 37˚C to 10.5 ± 0.3 min at 75˚C. Incorporating sputtered micro mirrors into the chip design increased the effective light path length, and improved signal-to-noise ratio 9-fold. The speed and reproducibility of the T50 chip-based assay run at 75˚C suggest that it may be suitable for rapid measurements, preferably in-line in a dialyser or in a portable microfluidic analytic device with the chip inserted as a disposable cartridge.

## Introduction

Chronic Kidney Disease (CKD) according to KDIGO (Kidney Disease: Improving Global Outcomes) guidelines requires that a measured glomerular filtration rate abnormality or evidence of kidney damage (e.g. albuminuria), or both, be present for a minimum of three months [1].

the Federal Republic of Germany (https://www.bmwi.de/Redaktion/EN/Artikel/SME-Sector/technology-neutral-project-support-01.html; grant number ZIM ZF4429601AJ7) awarded to WKS and by grants awarded to WJD by the IZKF Aachen of the Medical Faculty of RWTH Aachen (https://www.izkf.rwth-aachen.de/index.php/en/izkf-aachen) and by the German Research Foundation (https://www.dfg.de/en/index.jsp; DFG SFB/TRR219-Project C-03). The funders had no role in study design, data collection and analysis, decision to publish, or preparation of the manuscript. AP and WJD are co-founders and share holders of Calcisicon AG, a start-up company based at Nidau, Switzerland. AP is an employee of Calcisicon AG, who provided support in the form of salaries for author AP, but did not have any additional role in the study design, data collection and analysis, decision to publish, or preparation of the manuscript. The specific roles of these authors are articulated in the 'author contributions' section.

**Competing interests:** The authors have read the journal's policy and have the following conflicts: AP and WJD are co-founders and share holders of Calcisicon AG, a start-up company based at Nidau, Switzerland. AP is an employee of Calcisicon AG, who provided support in the form of salaries for author AP. The commercial affiliation does not alter our adherence to PLOS ONE policies on sharing data and materials.

End stage renal disease (ESRD) requires renal replacement therapy (dialysis or kidney transplantation). Phosphate retention in CKD and ESRD is a major driver of endothelial damage, and cardiovascular morbidity and mortality [2]. A disturbed phosphate homeostasis is closely associated with soft tissue calcifications and accelerated aging [3]. We and others have suggested that calciprotein particles (CPP), colloidal blood-borne particles containing calcium phosphate and plasma proteins, e.g. the hepatic glycoprotein fetuin-A and albumin, stabilize extracellular fluids supersaturated with calcium and phosphate, and are associated with CKD [4–12]. In CKD, hyperphosphatemia is the driving force of CPP formation [13–17], but calcium overload seems to be causing inflammation-associated tissue damage and calcification. CPP carry excess calcium and phosphate as colloids stabilized by plasma-derived "mineral chaperone proteins" [18]. In CKD, CPP are continuously formed, yet insufficiently cleared, and therefore seem to be the "culprit of phosphorous woes" [17]. Because serum phosphate is a well-established risk factor for CKD associated morbidity [3, 13–15], serum phosphate reduction is a major goal of dialysis apart from body fluid reduction and uremic toxin removal. However, the simple measurement of serum phosphate often does not correlate well with clinical outcome in patient cohorts, because it fails to detect the contribution of low and high molecular weight inhibitors of phosphate crystallization. We developed a functional test measuring the overall calcification propensity in blood serum or plasma [19]. This so called T50 test measures the time between mixing blood serum with calcium and phosphate, and the time at which colloidal mineral complexes called primary CPP have undergone half-maximum transformation to larger secondary CPP, hence the name T50 test.

The CPP transformation is detected as a sharp increase in turbidity of a test solution, which is measured by light scattering in a nephelometer. Serum samples of healthy subjects transform later than serum samples from dialysis patients. The test has been widely used to assess patient cohorts for prognosis and outcome including CKD patients [20, 21], hemodialysis patients [22, 23], and kidney transplant patients [24, 25]. The T50 test is a functional global assay measuring CPP transformation, which is influenced by low molecular regulators of mineralization like pyrophosphate and magnesium as well as large molecular weight plasma-derived mineral chaperone proteins [18]. Thus, CPP reflect better than serum phosphate "the culprit of phosphorous woes", i.e. phosphate toxicity [17, 26], especially if they are formed in excess or insufficiently cleared.

Variations of the original T50 test [27] have been employed for cross-sectional studies and for drug development [7, 28, 29]. The nephelometer-based assay requires measurement times of up to 600 minutes, rendering the test unsuitable for bed side measurements. Yet a rapid functional T50 test is highly desirable for point-of-care or even in-line measurements. Studying the kinetics of CPP formation and ripening we showed that particle ripening follows Arrhenius law [30]. Therfor the reaction kinetics should be logarithmically accelerated with increased temperature. In addition, we attempted to minimize test volumes to avoid reagent mixing and thermal issues to ultimately allow integration of the test in-line into existing dialysis equipment. Here we report that running the T50 test increasing the assay temperature from 37˚C to 75˚C reduced the T50 test time 36-fold from 381 ± 10 min at 37˚C to 10.5 ± 0.3 min at 75˚C. Incorporating sputtered micro mirrors into the chip design increased the effective light path length, and improved signal-to-noise ratio 9-fold.

## Material and methods

### Microfluidic chip fabrication

An aluminium replica tool for embossing the chip base was micro milled with a computer controlled micro mill (model M7 HP, Datron, Mühltal, Germany). The base of the microfluidic

chip was fabricated by ultrasonic hot embossing a stack of a 4 mm thick polycarbonate sheet (Arla Plast AB, Borensberg, Sweden) and a 125 μm thick polycarbonate foil (Rachow Kunststoff-Folien, Hamburg, Germany) using an ultrasonic welding machine (2000IW, Branson, Danbury, USA) as described [31, 32]. The base of the chip containing the measuring cell and microfluidic channels was ultrasonic hot embossed and 0.5 mm thick polycarbonate sheet was ultrasonically welded as a lid on top using the parameters listed in Table 1.

The aluminium replica tool containing microfluidic channels and energy directors was micro milled. For the integration of mirrors into the measuring cuvette, the edges of the cuvette wall were sloped using a 45˚ angle milling head. Mirrors were generated by sputtering a 170 nm layer of titanium onto the microchips after ultrasonic hot embossing and before ultrasonic welding the lid foil. To avoid overheating of the polymer, the titanium was sputtered in 6 steps, 10 s each, allowing intermittent intervals of 30 s for cooldown. An AZ 400 sputtering machine (Leybold Heraeus, Köln, Germany) was used at an argon flow of 100 sccm and a supply power of 400 W.

## Chip holder with Peltier element and temperature control

The chips were clamped into a holder. The chip temperature was adjusted by a Peltier element [32]. The ambient temperature and the Peltier element temperature were measured with PT1000 sensors using a LabVIEW interface program. Pre-mixed reaction fluid was injected by a syringe pump (KR Analytical, Cheshire, United Kingdom).

## Human serum samples

The use of human serum samples was approved by the local Ethical Board of RWTH Aachen University Clinics Ethical Board (www.medizin.rwth-aachen.de/EK; EK 300/14). Samples were from a biobank. Serum samples were from adult healthy donors and from dialysis patients who gave written informed consent. All samples used in this study were analyzed anonymously. Pooled anonymized serum samples were employed for the replica measurements shown in Fig 6.

## Reaction mixture and setup

The reaction mixture was prepared at room temperature in 1.5 ml reaction tubes by thoroughly mixing 0.4 mL pooled human serum and 0.35 mL of a 28.6 mmol $CaCl_2$ solution. Next, 0.25 mL of a 24 mmol $Na_2HPO_4$ solution was added and again thoroughly vortexed. Solutes and final concentrations were as published [33]. The reaction mixture was immediately injected into the inlet of the microfluidic chip using a Luer lock syringe attached to the inlet of the chip. The chip was clamped to the holder, illuminated with a laser diode (IMM-1255FB-635-1-E-G-L, IMM Photonics GmbH, Unterschleißheim, Deutschland) at 635 nm (Laser diode, 3.4×0.8 mm$^2$ beam cross section, 20 mm focus length), and turbidity in the light path

**Table 1. Parameters of ultrasonic processing.**

| Process | Hot embossing | Welding |
|---|---|---|
| Force threshold [N] | 290 | 170 |
| Amplitude of vibration [μm] | 27.2 | 24 |
| Force during vibrations [N] | 912 | 450 |
| Duration of vibrations [s] | 1.35 | 0.28 |
| Holding time [s] | 3 | 1.6 |
| Force during holding [N] | 912 | 450 |

was continuously monitored by a light dependent resistor (GM5528, Φ 5 Series, Wodeyijia Technology, Shenzhen, China).

A LabVIEW script controlled both the temperature and the voltage measurement at the light dependent resistor (LDR). All recorded data were stored in a log file that was further processed with Microsoft Excel.

## Results and discussion

### Microfluidic chip fabrication

The polymer chips were manufactured by ultrasonic hot embossing and welding of polycarbonate. Fig 1 shows the typical workflow of microfluidic chip production. During ultrasonic hot embossing, ultrasonic vibrations are generated by a sonotrode generating friction heat between the polymer base plate of the chip and a polymer foil, which was placed against the embossing aluminium tool (Fig 1A). The friction heat generated by the vibrations at the interface of tool and polymer foil and especially between polymer foil and polymer base plate melted the polymer (Fig 1B). The compressive force of the anvil replicated the micro structures of the aluminium tool. After a few seconds cooldown time the solidified chip base was demoulded (Fig 1C). The two layers of polymer (base plate and foil) were fused into a single piece of polymer. Fig 1D and 1E illustrate how the finished chip base was covered using a polymer lid, which was sealed around the edges of the chip microchannels along the energy directors. Fig 1F illustrates the release of the finished microchip. Fig 1G shows the actual micro machined aluminium tool used for embossing chips for transmitted laser light measurements. Fig 1I shows a matching polymer chip base plate with machined recesses, into which excess polymer can overflow during hot embossing. Fig 1H shows a finished hot embossed chip with the cover lid, but without connectors.

For embossing large grooves into the polymer, e.g. a sample cell with a depth of 1 mm and an area of 35 mm$^2$, a recess was machined into the polymer base plates before ultrasonic hot embossing. Thus, less molten polymer needed to be displaced and less vibration energy was required.

Ultrasonic movement of the polymer base plate against the polymer foil generated friction at the interface of polymer foil and tool, and also between polymer foil and base plate thus enhancing heat generation inside the polymer stack. As an added advantage, the melt between the polymer foil and the polymer base plate was thermally insulated from the cool and thermally conductive tool, facilitating polymer flow.

The microfluidic chips manufactured for this work were embossed using a combination of a 125 μm thick polycarbonate foil and a 4 mm thick polycarbonate base plate. The resulting chips were covered with a 0.5 mm thick polycarbonate lid. Ultrasonic welding was similar to embossing the base plate, but reduced vibration amplitude and force were selected to avoid damaging the micro channels. Instead, only the so-called energy directors melted as intended (Fig 1D–1F). Fabrication parameters are listed in Table 1.

Fig 2 shows a schematic of the microfluidic chip design next to the finished hot embossed real chips. Fig 2A represents a chip with translucent cuvette for transmitted laser light measurements.

Fig 2B shows an improved design featuring mirrored cuvette surfaces for reflected laser light measurement. This was achieved by inserting mirrors tilted by 45˚ relative to the bottom of the cuvette. In this chip, the laser light was reflected twice and its path through the sample solution was thus extended to 10 mm from 1 mm, the vertical dimension of the cuvette. Due to the longer path, more light was absorbed by the reaction mixture and a stronger signal was recorded by the LDR when primary CPP transformed into secondary CPP.

## Data acquisition and processing

The light sensitive detector had its highest sensitivity at low luminosity. Therefore, the laser diode was driven at a comparatively small electrical current of 10 mA. In the improved chip design shown in Fig 2B, a diffusor made of polypropylene (PP) was placed in front of the LDR, further reducing the light intensity per area of the LDR.

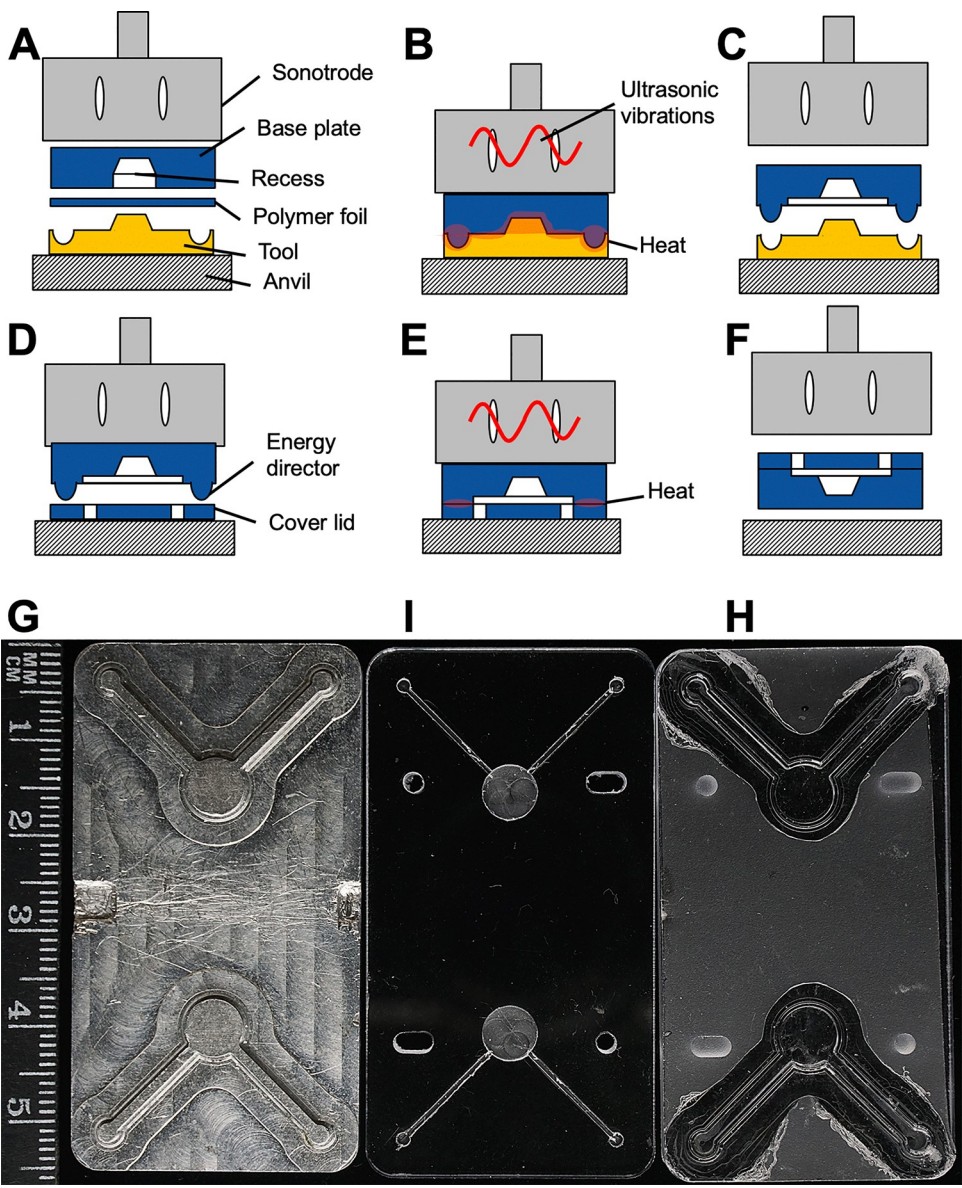

**Fig 1. Fabrication of a microchip by ultrasonic hot embossing and welding. A,** Polymer plate and foil are placed between tool and sonotrode; **B,** The sonotrode presses the polymer layers onto the tool and generates friction heat, the polymer layers are welded together to form a micro patterned chip base; **C,** After cooling down and solidifying, the chip base is removed from the tool; **D,** The micro patterned plate and a polymer lid are placed onto the anvil; **E,** The sonotrode welds both parts by friction heat; **F,** The sealed microfluidic chip is removed from the machine; **G,** Aluminum replica tool for two microfluidic chambers fabricated by micro milling; **I,** Polymer base plate with machined recesses for excess molten polymer; **H,** Ultrasonically hot embossed chip with cover lid.

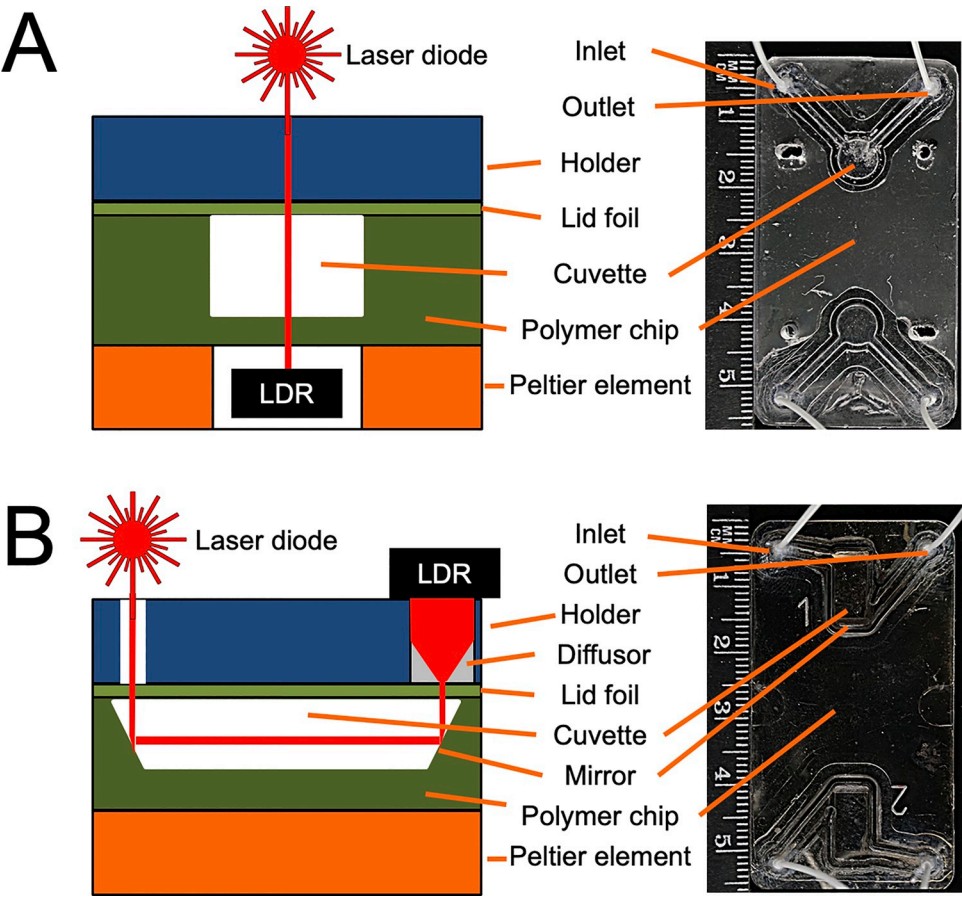

**Fig 2. Microfluidic polymer chips for calcium phosphate crystallization analyses produced by ultrasonic hot embossing and welding.** Each chip contains two cuvettes allowing two analyses in one chip. The drawing of the cross-sections is not to scale. **A,** A two chamber chip design with embossed fluidic channels and cuvette, a polymer foil lid designed for transmitted laser light measurements. **B,** An improved design including sputtered titanium mirrors on either side of the cuvette, and a diffusor in front of the detector, effectively increasing the measuring path length and distributing the laser light transmitted through the cuvette over the entire surface of the light dependent resistor, LDR.

Fig 3A shows a plot of the light intensity as a function of time measured with the light dependent sensor and the microchip shown in Fig 2A. The curve was smoothed by plotting the sliding average of the voltages measured 100 s before and 100 s after each data point. The time range of 100 s was chosen because it yielded a smoothed curve without flattening the overall curve.

In an alternative plot shown in Fig 4B, the difference of the average of voltages 200 s after each time point and the average of the 200 s before this time point were plotted against time. This representation is similar to the first derivative of the turbidity. The maximum of the derivative corresponds to the time of the maximum slope of the turbidity curve shown in Fig 3A and represents the T50 time. The curve peak was determined by a $\chi^2$-fit resulting in the parabola plotted as an orange dotted line in Fig 3B. In this representation, T50 corresponds to the vertex of the parabola determined by the $\chi^2$-fit. This way, the maximum slope of the curve in Fig 3A is not based on a single data point subject to scatter but includes all the measured turbidity changes in its vicinity, resulting in better accuracy.

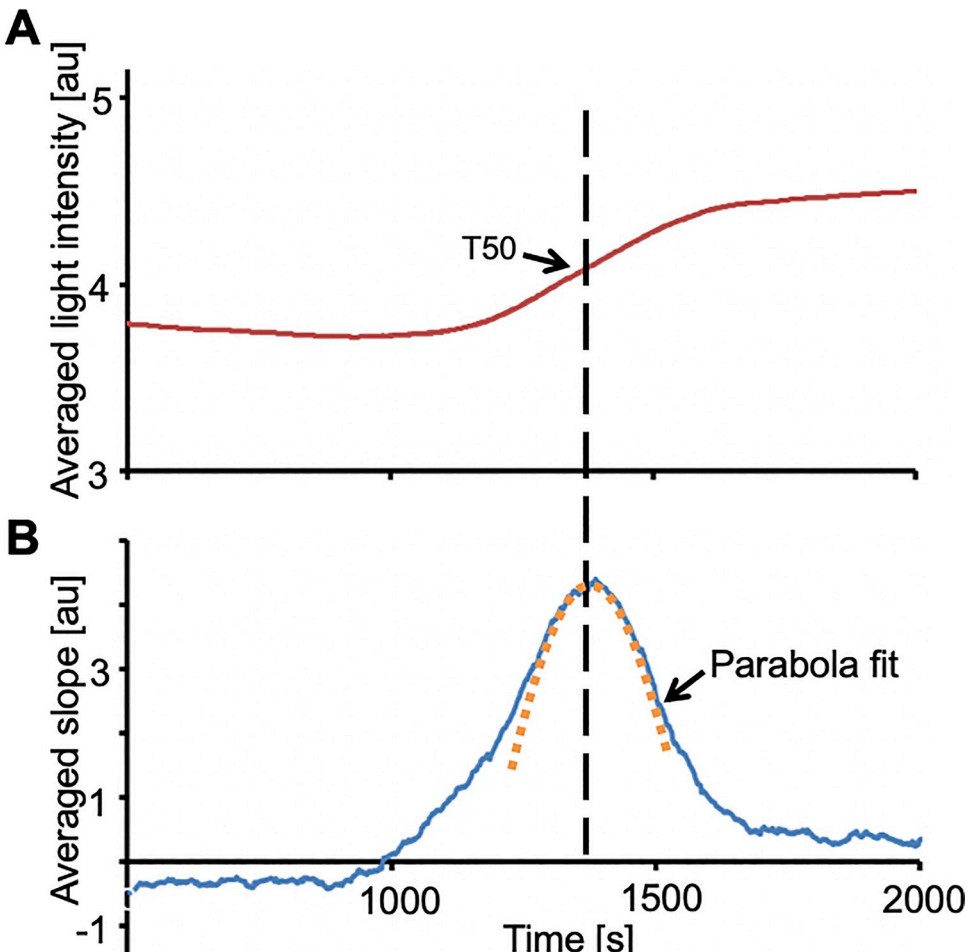

**Fig 3. Determination of T50-value by a parabola fit to the first order derivative of light intensity transmitted through the sample. A,** The red line represents the smoothed light intensity plotted against time. **B,** The blue line corresponds to the first order derivative of the light intensities. T50 corresponds to the vertex of the parabola determined by a $\chi^2$-fit, which is shown as an orange dotted line.

Fig 4 shows T50 measurements taken with identical serum at 65˚C with and without mirrors built into the chip (blue line and red line, respectively). The integration of mirrors increased the effective length of the light path from 1 mm to roughly 10 mm as illustrated in Fig 2. The voltage increase used to determine the T50 value was 1.8 V and 0.2 V with and without mirrors, respectively, demonstrating a 9-fold increase in sensitivity.

Fig 4A shows raw data with spikes, and Fig 4B shows data representing the sliding average of 40 s before and after each time. Notably, both Fig 4A and Fig 4B show that the signal to noise ratio and the overall voltage step were much improved by inserting mirrors, resulting in an overall more robust determination of T50.

## Results of the miniaturized calcification propensity test

Human serum was mixed with calcium and phosphate, and the T50 times were measured at temperatures ranging from 60 to 85˚C employing microchips with integrated mirrors. The change in turbidity due to the formation of secondary CPPs was detected as a function of time.

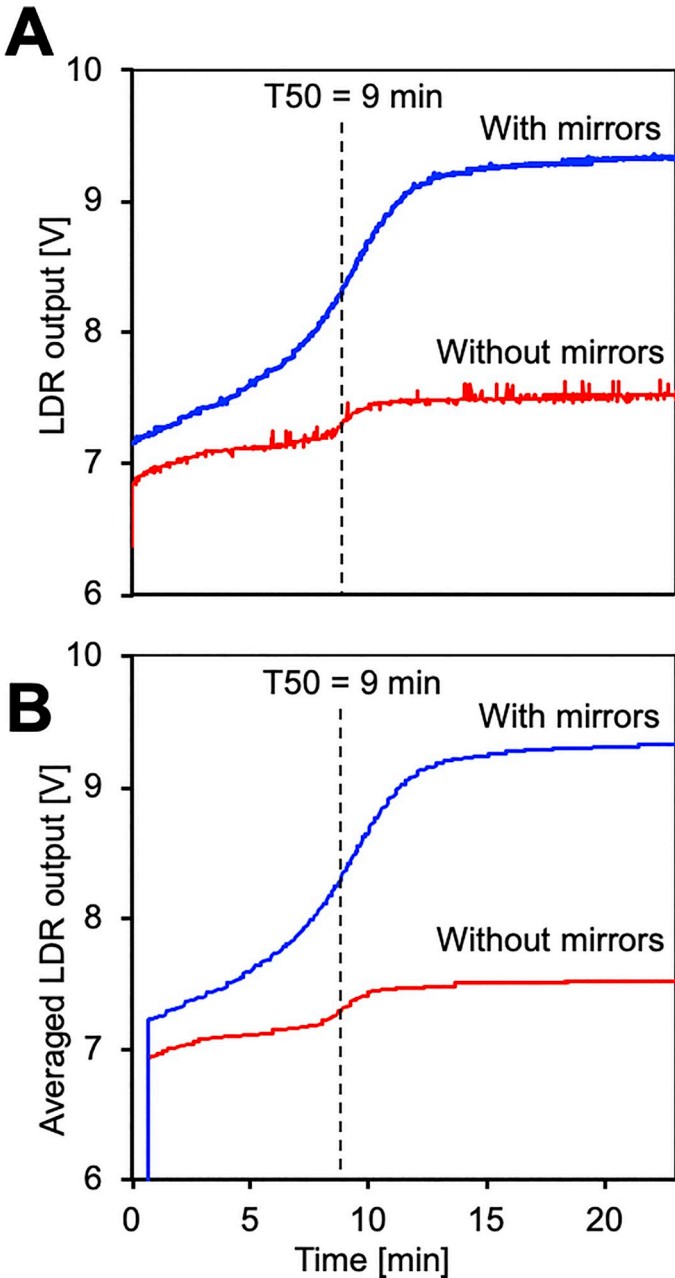

**Fig 4. T50 measurement with and without mirrors in the microchip.** T50 measurements of identical serum samples were taken at 65˚C. **A,** Direct light dependent resistor (LDR) output, **B,** sliding average LDR output results in curve smoothing. Mirrors in the microchip increased the effective length of the measuring path through the sample cell from 1mm to 10 mm, reduced noise and increased the overall voltage step.

Measurements at each temperature were done once with identical serum, each with a fresh chip.

Fig 5 shows that increasing the assay temperature strongly reduced T50 time. At 60˚C a swift increase in turbidity within the first 5 minutes suggested protein denaturation, which was followed by the typical sigmoidal curve indicating transformation of primary CPP into secondary CPP at T50 = 34 minutes. Increasing the assay temperature beyond 60˚C cut off the

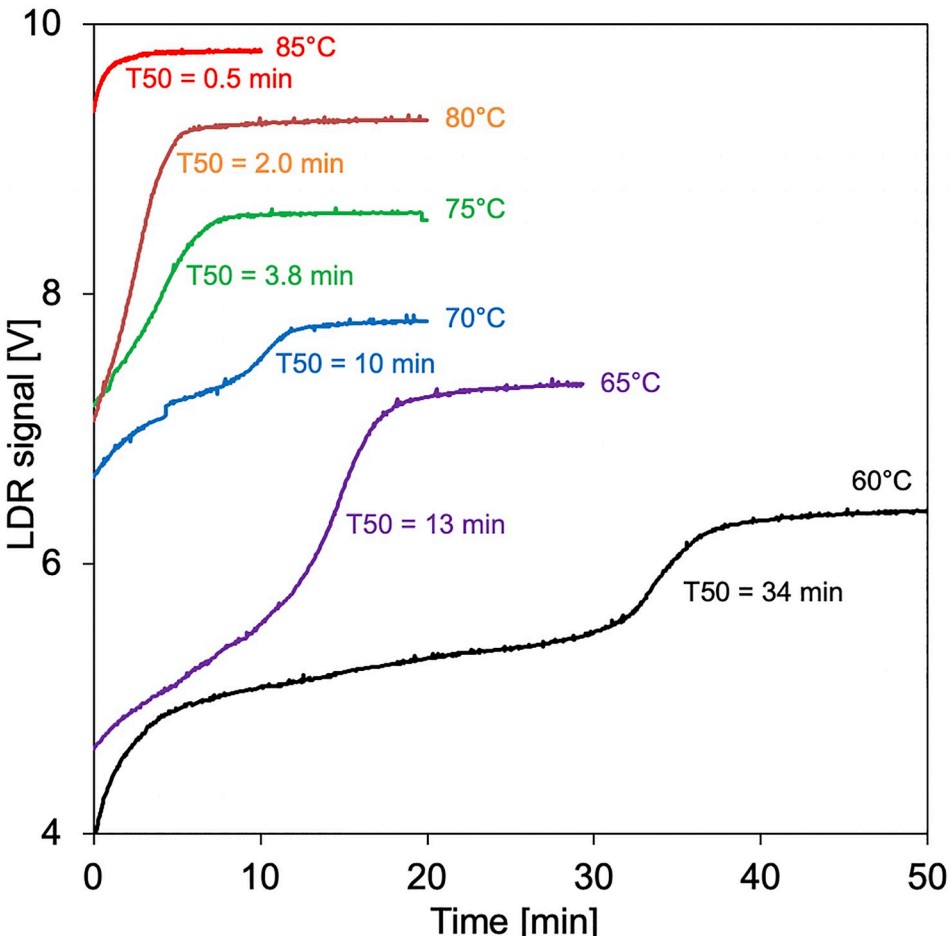

**Fig 5. T50 chip tests at elevated temperature.** Curves are staggered along the y-axis for clarity.

initial protein denaturation signal, and further reduced the T50 times to 13, 10, 3.8, 2 and 0.5 minutes at temperatures 65, 70, 75, 80 and 85˚C, respectively. Raising the temperature beyond 75˚C reduced reproducibility at even the slightest variations in pipetting and handling. Thus, we choose 75˚C for a comparison of chip assay-derived T50 values with nephelometer assay-derived T50 values (Fig 6).

All nephelometer measurements were repeated 5-fold, and every chip-based measurement was repeated 15-fold. The mean coefficient of variation (standard deviation divided by mean value) derived from microchip measurements (5.7% in the average of all measurements) was smaller than the one derived from nephelometer measurements (7.9%) suggesting excellent reproducibility of both chip manufacturing and test performance. Fig 6 shows linear correlation ($R^2$ = 0.9892) of nephelometer and chip-derived T50 values. The close correlation of both measurement routines indicates that the T50 values truly reflected the calcification propensity of the serum samples and not the rate of protein denaturation, which would have been expected to be identical in all sera, because they contained similar amounts of total protein. The measuring range presented in Fig 6 covered previously published nephelometer-based T50 values including a reference range of healthy subjects (270 to 460 minutes), determined in 253 Swiss adults [21], values determined in 184 patients with CKD stages 3 to 4 (mean,

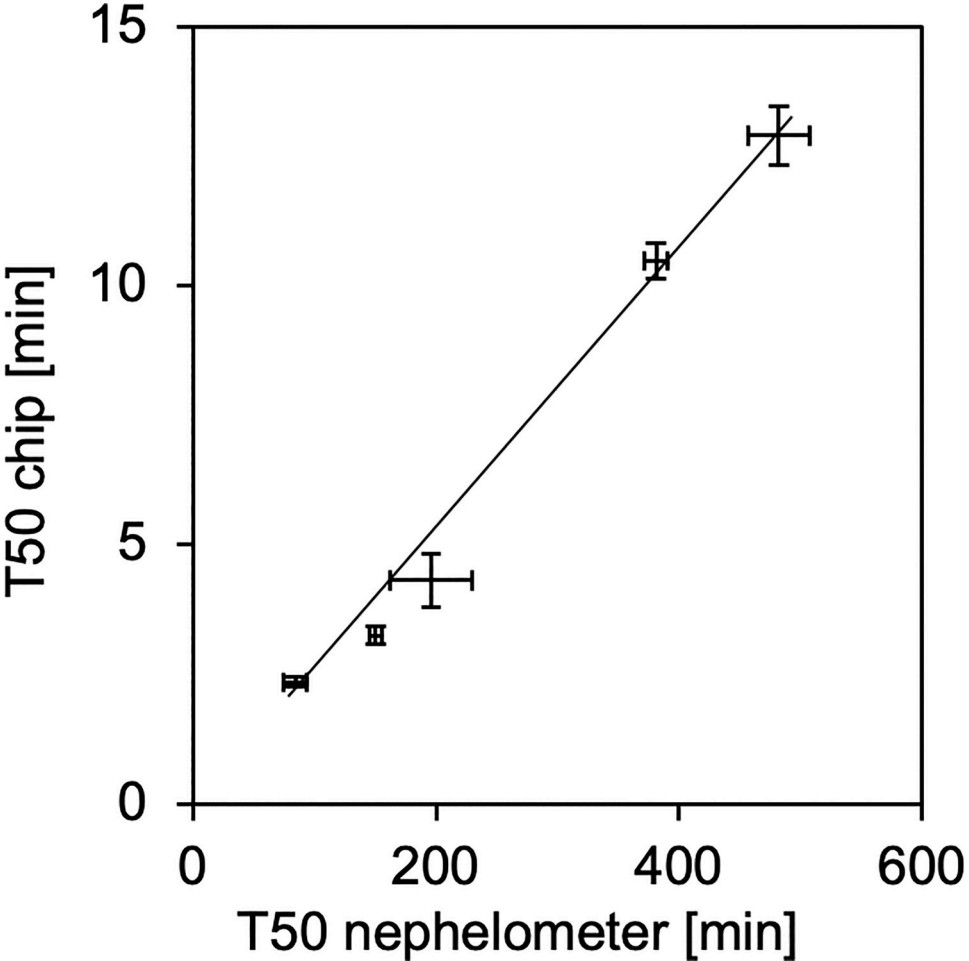

**Fig 6. Serum T50 values derived from chip measurements at 75˚C compared to nephelometer measurements at 37˚C.** The error bars denote the standard errors of 15 and 5 independent replicate measurements in chip (75˚C) and nephelometer assay (37˚C), respectively, in five human serum pools covering the entire T50 range observed in published clinical studies.

329 ± 95 minutes) [20] and in 2,785 patients undergoing hemodialysis (median, 212 [10th-90th percentiles, 109–328] minutes) [22].

## Discussion and conclusions

We present a proof-of-principle miniaturized microfluidic T50 test format reporting overall calcification propensity in serum. The test has improved sensitivity and speed over the existing nephelometer-based test, which is the current gold standard. Before the chip-based test can be evaluated in large cohorts like the nephelometer-based test, an integrated cartridge holder with temperature and fluidics control unit, and data analysis must be developed. It should be noted in favour of the nephelometer-based test that automated reaction mixing was part of this test, but not of the chip-based test, and therefore the overall performance may not be strictly comparable. Nevertheless, our work indicates that the assay can be accelerated from hours to minutes by increasing the assay temperature, which is impossible using the established microtiter plate nephelometer-based assay. Short analysis times are advantageous in clinical settings when T50 measurements inform about the completeness of dialysis or about the influence of

varying dialysis protocols or fluids. Importantly, the T50 times reflect the combined calcification propensity including solutes, activators and inhibitors as well as CPP, which might serve as mineralization nuclei that are poorly reflected by conventional phosphate monitoring. Recently, CPP and their precursors were directly end elegantly measured by two independent methods [6, 34]. These and similar methods may further refine the assessment of calcification propensity if miniaturized and accelerated like the chip-based T50 assay presented here.

The T50 acceleration was not linear, which might have been expected given that previous time-resolved small-angle X-ray and quantitative small-angle neutron scattering studies indicated first order CPP formation kinetics at 37°C [35–37]. We hypothesize that CPP formation and transformation of primary into secondary CPP at elevated temperature are more complex, because formation and dissolution of primary CPP, pH stability of the buffer system, and thermal denaturation of serum proteins will all influence the overall T50 time apart form the solute concentrations. These points require further physicochemical studies beyond the scope of this work. Nevertheless, we report close correlation of test temperature and T50 time up to 75°C. This was somewhat unexpected, because this temperature is above the denaturation temperature of many proteins. Published work shows however, that many serum proteins and complex protein solutions withstand high temperatures. Serum for cell culture is commonly heated to 56°C to inactivate complement proteins without affecting growth promoting and attachment factors. Proteins from thermophilic bacteria withstand temperatures above 100°C, e.g. the widely employed polymerase chain reaction enzyme Taq polymerase. Fetuin-A and albumin, two major protein components of CPP [11, 38] are also remarkably stable against thermal denaturation due to multiple internal disulphide bridges stabilizing their three-dimensional structure. Both proteins withstand temperatures beyond 70°C [39, 40] explaining why the chip-based T50 assay worked even at this high temperature. The proteins involved in CPP formation likely gain further stability to heat denaturation, because they are associated with a calcium phosphate solid phase.

Using a temperature controlled microfluidic polymer chip and serum samples from volunteers, we studied systematically the temperature dependent T50 assay acceleration and compared results with the established nephelometer test. We observed 36-fold acceleration and excellent correlation of results obtained from the established nephelometer test run at 37°C with results obtained with a temperature controlled microfluidic polymer chip operated at up to 75°C.

In conclusion, the speed and reproducibility of the T50 chip-based assay run at 75°C suggest that it may be suitable for rapid measurements, preferably in-line in a dialyser or in a portable microfluidic analytic device with the chip inserted as a disposable cartridge.

## Acknowledgments

A pre-publication manuscript of this work was deposited as a printed abstract of the Biomedical Technology Conference BMT2018 [41].

## Author Contributions

**Conceptualization:** Andreas Pasch, Werner Karl Schomburg, Willi Jahnen-Dechent.

**Data curation:** Julia Bavendiek, Andreas Pasch, Werner Karl Schomburg.

**Formal analysis:** Julia Bavendiek.

**Funding acquisition:** Werner Karl Schomburg, Willi Jahnen-Dechent.

**Investigation:** Julia Bavendiek, Philip Maurer, Steffen Gräber, Andreas Pasch.

**Methodology:** Julia Bavendiek, Steffen Gräber, Andreas Pasch.

**Project administration:** Werner Karl Schomburg.

**Software:** Julia Bavendiek, Philip Maurer.

**Supervision:** Werner Karl Schomburg, Willi Jahnen-Dechent.

**Visualization:** Julia Bavendiek, Willi Jahnen-Dechent.

**Writing – original draft:** Julia Bavendiek, Philip Maurer, Willi Jahnen-Dechent.

**Writing – review & editing:** Julia Bavendiek, Andreas Pasch, Werner Karl Schomburg, Willi Jahnen-Dechent.

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
