## [Decision Letter · Decision Letter 0]

8 Nov 2019

PONE-D-19-22409

Rapid calcification propensity testing in blood using a temperature controlled microfluidic polymer chip

PLOS ONE

Dear Prof Jahnen-Dechent,

Thank you for submitting your manuscript to PLOS ONE. After careful consideration, we feel that it has merit but does not fully meet PLOS ONE’s publication criteria as it currently stands. Therefore, we invite you to submit a revised version of the manuscript that thoroughly addresses all points raised during the review process and quoted below.

We would appreciate receiving your revised manuscript by Dec 23 2019 11:59PM. To enhance the reproducibility of your results, we recommend that if applicable you deposit your laboratory protocols in protocols.io, where a protocol can be assigned its own identifier (DOI) such that it can be cited independently in the future. For instructions see: http://journals.plos.org/plosone/s/submission-guidelines#loc-laboratory-protocols

We look forward to receiving your revised manuscript.

Kind regards,

Marc W. Merx, MD

Academic Editor

PLOS ONE

Journal Requirements:

2. Thank you for including your ethics statement: RWTH Aachen University Clinics Ethical Board (www.medizin.rwth-aachen.de/EK; EK

300/14), data were analyzed anonymously  

Please add your ethics statement to the Methods section of the manuscript and ensure you specify that the named ethics committee specifically approved your study

3. Please provide additional details about the human serum samples used in this study in the Methods section of your manuscript. If the serum samples were obtained from a biobank or blood bank, please state this in the Methods. However, if the serum samples were drawn from human participants for the purpose of this work, please specify the following information;

- The dates (month/year) that the serum samples were collected

- The number of donors and how they were recruited

- Whether you obtained participant consent. Please ensure that you have specified whether (1) consent was informed and (2) what type you obtained (for instance, written or verbal, and if verbal, how it was documented and witnessed). If your study included minors, state whether you obtained consent from parents or guardians. If the need for consent was waived, please ensure that you have discussed whether all data were fully anonymized before you accessed them and/or whether the IRB or ethics committee waived the requirement for informed consent.

Thank you for your attention to these requests.

AP and WJD are co-founders and share holders of Calciscon AG, a start-up company based at Nidau, Switzerland. AP is an employee of Calciscon AG, who provided support in the form of salaries for authors AP, but did not have any additional role in the study design, data collection and analysis, decision to publish, or preparation of the manuscript. The commercial affiliation does not alter our adherence to PLoS ONE policies on sharing data and materials.

We note that one or more of the authors are employed by a commercial company: Calciscon AG

Reviewers' comments:

Reviewer's Responses to Questions

**Comments to the Author**

1. Is the manuscript technically sound, and do the data support the conclusions?

Reviewer #1: Yes

Reviewer #2: Yes

2. Has the statistical analysis been performed appropriately and rigorously? 

Reviewer #1: Yes

Reviewer #2: Yes

3. Have the authors made all data underlying the findings in their manuscript fully available?

Reviewer #1: Yes

Reviewer #2: Yes

4. Is the manuscript presented in an intelligible fashion and written in standard English?

Reviewer #1: Yes

Reviewer #2: No

5. Review Comments to the Author

Reviewer #1: In this manuscript, Bavendiek et al. present a newly designed microfluidic polymer chip which reduces times for measuring calcification propensity and may form a basis for future point of care tests. The most important part of this test is that measurement of calcification propensity can be achieved at much higher temperatures than the previous test thereby reducing measurement time from 600 to just a few minutes.

It is a engineering-orientated paper with detailed reports on the fabrication and optimization of the microchip.

Main comments:

- What would be the benefit of a possible bedside test? Would there be changes in conducting dialysis or is calcification propensity rather a long-term parameter?

- The introduction does not match the actual paper. Introduction handles kidney disease, hyperphosphatemia, morbidity, dialysis, clinical outcome etc, but the paper is an engineering-based approach. This should be adjusted to either more data on clinical use (e.g. adding a small clinical trial) or going all the way to an engineering paper.

- The sera of 5 patients was used for validation of the test. But there is no data on the used test sera (Figure 7). Did the patients suffer from clinically relevant calcification? CAD?

- Figure 1 seems to be redundant.

Reviewer #2: Bavendiek et al demonstrated an improved temperature controlled T50 test format which reports Overall calcification propensity in serum.

While this is an important topic, there are some major limitations of this manuscript which might it challenging to follow. The authors should more emphasize the aim of the study, shorten the introduction and extend the discussion.

6. PLOS authors have the option to publish the peer review history of their article (what does this mean?). If published, this will include your full peer review and any attached files.

Reviewer #1: No

Reviewer #2: No

---

## [Author Response · Author response to Decision Letter 0]

20 Dec 2019

Response To Reviewers

Our answer: We would like to thank all reviewers for their time and effort to improve the quality of our manuscript. We respond to their comments point-by-point.

Reviewers' comments:

Reviewer's Responses to Questions

Comments to the Author

1. Is the manuscript technically sound, and do the data support the conclusions?

Reviewer #1: Yes

Reviewer #2: Yes

2. Has the statistical analysis been performed appropriately and rigorously? 

Reviewer #1: Yes

Reviewer #2: Yes

3. Have the authors made all data underlying the findings in their manuscript fully available?

Reviewer #1: Yes

Reviewer #2: Yes

4. Is the manuscript presented in an intelligible fashion and written in standard English?

Reviewer #1: Yes

Reviewer #2: No

5. Review Comments to the Author

Reviewer #1: In this manuscript, Bavendiek et al. present a newly designed microfluidic polymer chip which reduces times for measuring calcification propensity and may form a basis for future point of care tests. The most important part of this test is that measurement of calcification propensity can be achieved at much higher temperatures than the previous test thereby reducing measurement time from 600 to just a few minutes.

It is a engineering-orientated paper with detailed reports on the fabrication and optimization of the microchip.

Our answer: We thank reviewer 1 for their appreciation of our work and the suggestions made in the comments.

Main comments:

- What would be the benefit of a possible bedside test? Would there be changes in conducting dialysis or is calcification propensity rather a long-term parameter?

Our answer: The benefit of a bedside, or better still, an in-line test could be patient individualized dialysis. An example may be taken from work cited in ref. 30 showing the “The Effect of Increasing Dialysate Magnesium on Serum Calcification Propensity in Subjects with End Stage Kidney Disease”. While such patient-individualized dialysis may still seem a long way off, an improved T50 test could be an important first step, because it better reflects the phosphorous woes (see ref 17) comprising phosphate, serum protein concentration, magnesium, pyrophosphate etc. than does serum phosphate alone.

- The introduction does not match the actual paper. Introduction handles kidney disease, hyperphosphatemia, morbidity, dialysis, clinical outcome etc, but the paper is an engineering-based approach. This should be adjusted to either more data on clinical use (e.g. adding a small clinical trial) or going all the way to an engineering paper.

Our answer: We thank the reviewer for this suggestion. In the introduction we toned done on clinical findings and focused on the task at hand, technical improvement of the T50 assay.

- The sera of 5 patients was used for validation of the test. But there is no data on the used test sera (Figure 7). Did the patients suffer from clinically relevant calcification? CAD?

Our answer: We have no information on the health status of the donors, because the samples were anonymized. To perform the repeat measurements shown in Fig 6 (was Fig 7) anonymized serum samples were blended to cover the range of published T50 studies, and to obtain sufficient amount of serum for repeat measurements. For clinical information we refer reviewer to the clinical studies employing the original T50 assay, some of which are cited.

- Figure 1 seems to be redundant.

Our answer: we deleted Fig 1.

Reviewer #2: Bavendiek et al demonstrated an improved temperature controlled T50 test format which reports Overall calcification propensity in serum.

While this is an important topic, there are some major limitations of this manuscript which might it challenging to follow. The authors should more emphasize the aim of the study, shorten the introduction and extend the discussion.

Our answer: We thank the reviewer for this suggestion. In the introduction we toned done on clinical findings and focused on the task at hand, technical improvement of the T50 assay. At this stage we would like to refrain from further discussion of this proof-of-principle study until we have completed a first clinical study using the novel chip-based T50 assay. We hope the reviewers can agree with this.

---

## [Decision Letter · Decision Letter 1]

27 Jan 2020

PONE-D-19-22409R1

Rapid calcification propensity testing in blood using a temperature controlled microfluidic polymer chip

PLOS ONE

Dear Prof Jahnen-Dechent,

Thank you for submitting your manuscript to PLOS ONE. After careful consideration, we feel that it has merit but does not fully meet PLOS ONE’s publication criteria as it currently stands. Therefore, we invite you to submit a revised version of the manuscript that addresses the points raised during the review process and quoted below.

We would appreciate receiving your revised manuscript by Mar 12 2020 11:59PM. To enhance the reproducibility of your results, we recommend that if applicable you deposit your laboratory protocols in protocols.io, where a protocol can be assigned its own identifier (DOI) such that it can be cited independently in the future. For instructions see: http://journals.plos.org/plosone/s/submission-guidelines#loc-laboratory-protocols

We look forward to receiving your revised manuscript.

Kind regards,

Marc W. Merx, MD

Academic Editor

PLOS ONE

Reviewers' comments:

Reviewer's Responses to Questions

**Comments to the Author**

1. If the authors have adequately addressed your comments raised in a previous round of review and you feel that this manuscript is now acceptable for publication, you may indicate that here to bypass the “Comments to the Author” section, enter your conflict of interest statement in the “Confidential to Editor” section, and submit your "Accept" recommendation.

Reviewer #1: (No Response)

Reviewer #2: (No Response)

2. Is the manuscript technically sound, and do the data support the conclusions?

Reviewer #1: Yes

Reviewer #2: Yes

3. Has the statistical analysis been performed appropriately and rigorously? 

Reviewer #1: Yes

Reviewer #2: Yes

4. Have the authors made all data underlying the findings in their manuscript fully available?

Reviewer #1: Yes

Reviewer #2: Yes

5. Is the manuscript presented in an intelligible fashion and written in standard English?

Reviewer #1: Yes

Reviewer #2: Yes

6. Review Comments to the Author

Reviewer #1: The authors present their revised manuscript on a newly designed microfluidic polymer chip which reduces times for measuring calcification propensity and may form a basis for future in-line tests. The most important part of this test is that measurement of calcification propensity can be achieved at much higher temperatures than the previous test thereby reducing measurement time from 600 to just a few minutes. It is an engineering-orientated paper with detailed reports on the fabrication and optimization of the microchip.

The manuscript has significantly improved due to the changes made in the introduction part.

Main comments:

- We suggest to include one sentence/paragraph on the impact of reduced CPP measurements on dialysis quality in discussion.

Reviewer #2: In the revised manuscript the authors have adequately addressed the reviewers' comments. I would recommend to add one phrase about the expected clinical benefit/ relevance into the abstract.

7. PLOS authors have the option to publish the peer review history of their article (what does this mean?). If published, this will include your full peer review and any attached files.

Reviewer #1: No

Reviewer #2: No

---

## [Author Response · Author response to Decision Letter 1]

28 Jan 2020

We thank both reviewers for their continued help and favorable judgement including their suggestion to comment on the potential clinical impact of this chip. We are happy to oblige. As before, we comment point-by-point quoting the original text. We hope the manuscript now meets the publication criteria.

Kind regards

Willi Jahnen-Dechent, on behalf of the authors

Comments to the Author

6. Review Comments to the Author

Reviewer #1: The authors present their revised manuscript on a newly designed microfluidic polymer chip which reduces times for measuring calcification propensity and may form a basis for future in-line tests. The most important part of this test is that measurement of calcification propensity can be achieved at much higher temperatures than the previous test thereby reducing measurement time from 600 to just a few minutes. It is an engineering-orientated paper with detailed reports on the fabrication and optimization of the microchip.

The manuscript has significantly improved due to the changes made in the introduction part.

Main comments:

- We suggest to include one sentence/paragraph on the impact of reduced CPP measurements on dialysis quality in discussion.

Our comment: Following reviewer’s suggestion we added a short paragraph to the discussion

“Short analysis times are advantageous in clinical settings when T50 measurements inform about the completeness of dialysis or about the influence of varying dialysis protocols or fluids. Importantly, the T50 times reflect the combined calcification propensity including solutes, activators and inhibitors as well as CPP, which might serve as mineralization nuclei that are poorly reflected by conventional phosphate monitoring. Recently, CPP and their precursors were directly end elegantly measured by two independent methods [6, 34]. These and similar methods may further refine the assessment of calcification propensity if miniaturized and accelerated like the chip-based T50 assay presented here.”

Reviewer #2: In the revised manuscript the authors have adequately addressed the reviewers' comments. I would recommend to add one phrase about the expected clinical benefit/ relevance into the abstract.

Our comment: We added a sentence to the abstract

“The speed and reproducibility of the T50 chip-based assay run at 75°C suggest that it may be suitable for rapid measurements, preferably in-line in a dialyser or in a portable microfluidic analytic device with the chip inserted as a disposable cartridge.”

---

## [Decision Letter · Decision Letter 2]

3 Mar 2020

Rapid calcification propensity testing in blood using a temperature controlled microfluidic polymer chip

PONE-D-19-22409R2

Dear Dr. Jahnen-Dechent,

We are pleased to inform you that your manuscript has been judged scientifically suitable for publication and will be formally accepted for publication once it complies with all outstanding technical requirements.

With kind regards,

Marc W. Merx, MD

Academic Editor

PLOS ONE

Additional Editor Comments (optional):

Reviewers' comments:

Reviewer's Responses to Questions

**Comments to the Author**

1. If the authors have adequately addressed your comments raised in a previous round of review and you feel that this manuscript is now acceptable for publication, you may indicate that here to bypass the “Comments to the Author” section, enter your conflict of interest statement in the “Confidential to Editor” section, and submit your "Accept" recommendation.

Reviewer #1: All comments have been addressed

Reviewer #2: All comments have been addressed

2. Is the manuscript technically sound, and do the data support the conclusions?

Reviewer #1: Yes

Reviewer #2: Yes

3. Has the statistical analysis been performed appropriately and rigorously? 

Reviewer #1: Yes

Reviewer #2: Yes

4. Have the authors made all data underlying the findings in their manuscript fully available?

Reviewer #1: Yes

Reviewer #2: Yes

5. Is the manuscript presented in an intelligible fashion and written in standard English?

Reviewer #1: Yes

Reviewer #2: Yes

6. Review Comments to the Author

Reviewer #1: (No Response)

Reviewer #2: All reviewers' comments are adequately adressed. I would suggest to accept this manuscript for publication.

7. PLOS authors have the option to publish the peer review history of their article (what does this mean?). If published, this will include your full peer review and any attached files.

Reviewer #1: No

Reviewer #2: No

---

## [Editor Report · Acceptance letter]

5 Mar 2020

PONE-D-19-22409R2 

Rapid calcification propensity testing in blood using a temperature controlled microfluidic polymer chip 

Dear Dr. Jahnen-Dechent:

I am pleased to inform you that your manuscript has been deemed suitable for publication in PLOS ONE. Congratulations! Your manuscript is now with our production department. 

With kind regards,

on behalf of

Prof. Dr. Marc W. Merx 

Academic Editor

PLOS ONE